# From small-scale forest structure to Amazon-wide carbon estimates

Edna Rödig [1,2]*, Nikolai Knapp[1], Rico Fischer [1], Friedrich J. Bohn[1], Ralph Dubayah[3], Hao Tang [3] & Andreas Huth[1,4,5]

Tropical forests play an important role in the global carbon cycle. High-resolution remote sensing techniques, e.g., spaceborne lidar, can measure complex tropical forest structures, but it remains a challenge how to interpret such information for the assessment of forest biomass and productivity. Here, we develop an approach to estimate basal area, aboveground biomass and productivity within Amazonia by matching 770,000 GLAS lidar (ICESat) profiles with forest simulations considering spatial heterogeneous environmental and ecological conditions. This allows for deriving frequency distributions of key forest attributes for the entire Amazon. This detailed interpretation of remote sensing data improves estimates of forest attributes by 20–43% as compared to (conventional) estimates using mean canopy height. The inclusion of forest modeling has a high potential to close a missing link between remote sensing measurements and the 3D structure of forests, and may thereby improve continent-wide estimates of biomass and productivity.

[1] Department of Ecological Modelling, UFZ - Helmholtz Centre for Environmental Research, Permoserstr. 15, 04318 Leipzig, Germany. [2] Department of Computational Hydrosystems, UFZ - Helmholtz Centre for Environmental Research, Permoserstr. 15, 04318 Leipzig, Germany. [3] Department of Geographical Sciences, University of Maryland, College Park, 2120 Lefrak Hall, College Park, MD 20742, USA. [4] University of Osnabrück, Barbarastraße 12, 49076 Osnabrück, Germany. [5] German Centre for Integrative Biodiversity Research (iDiv) Halle-Jena-Leipzig, Deutscher Platz 5e, 04103 Leipzig, Germany. *email: edna.roedig@ufz.de

Tropical forests store a large amount of carbon[1]. Under-standing and predicting the spatial variability of tropical forest biomass is hence important for the assessment of the global carbon cycle[2]. The consideration of three-dimensional structural information of forests has proven to be essential for carbon stock estimates to reflect successional states and differences in forest structure such as tree height[3–5].

The lidar system Geoscience Laser Altimeter System (GLAS) onboard the Ice, Cloud and land Elevation Satellite (ICESat) has contributed to identifying variability of forest height metrics in space to derive several biomass maps of tropical forests[6–8] and complement traditional remote sensing techniques (passive optical instruments)[9,10] that monitor land-cover changes[11,12] and productivity of vegetation[13–16].

However, assessing forest biomass with lidar confronts several challenges. These arise, among others, from the fact that the interpretation of lidar remote sensing measurements is based on statistical relations (e.g., height to biomass) that are derived from field inventory data. Although nowadays valuable inventory data exist (e.g., ForestPlots[17]), those inventory data are rarely available at the same spatial extent and at the location of the lidar measurements, in particular, in large regions such as the Amazon rainforest. Second, field inventories often do not follow a systematic spatial sampling strategy to reveal landscape patterns[18]. Third, biomass maps derived from remote sensing mainly build on a general pan-topical relation between AGB and a lidar height metric although it is known that these can vary for different regions, e.g., due to variations in tree wood densities[19–21].

The combination of lidar measurements and forest modeling provides new possibilities[22,23]. It allows for including canopy height information from remote sensing, on the one hand, e.g.[24], and for considering regional differences in ecological characteristics (e.g., mortality, turn-over, wood density) as observed in the field, e.g.[25], on the other hand. In a recent study[26], a forest model has been applied under spatially variable environmental conditions on the entire Amazon rainforest in order to link simulated forests stands with a canopy height map[27] derived from GLAS lidar. The integration of forest modeling was a first step toward bridging a gap across different spatial scales of field inventories and lidar remote sensing. Nevertheless, it also implies uncertainties as one canopy height may be associated with different forest stands, including different species compositions and biomass values. Hence, this previous study considers upper canopy information but ignores sub-canopy structures[28]. Empirical remote-sensing studies have shown that various metrics derived from waveforms are needed to describe the structure of a forest[29] and, therefore, should be assimilated into ecosystem models[24].

Owing to increasing computational capacities, we are now able to explore the integration of entire canopy profiles within a forest model at large spatial scales, such as the Amazon rainforest. In this study, we use canopy profiles recorded by GLAS lidar[30] in order to assess the biomass distribution of the Amazon. The Amazon-wide version[26] of the forest model FORMIND[31,32] is used to simulate forest dynamics at the scale of a GLAS lidar shot (a circular area of 65 m diameter) from which we reconstruct full lidar profiles. We match ~770,000 lidar profiles in the Amazon (GLAS) with simulated profiles. For every lidar shot, we then derive probability distributions of potential forest attributes (here: aboveground biomass (AGB), stem volume (SV), basal area (BA), gross primary productivity (GPP), and aboveground woody productivity (AWP)). The approach is used to address the two primary questions: (1) How much information about forests (AGB, SV, BA, GPP, AWP) can be derived from full lidar profiles? (2) Can we reduce the uncertainties in estimates of forest attributes when using entire profiles as compared with using solely mean canopy height (MCH)? Among all forest attributes tested, we find the highest uncertainties for AGB estimations. The uncertainties decrease with increasing forest height. We further show that we can extract more information from entire profiles than just a single lidar metric.

## Results

**The uncertainty index $\varepsilon$.** Each GLAS lidar profile in the Amazon rainforest was compared with profiles derived from simulated forest succession to filter out the potential successional states of forests (Fig. 1). The best matches were then taken to estimate forest attributes, e.g., AGB. It is hence possible to derive a probability distribution for biomass at each ICESat shot (Fig. 2). The analysis shows that for some locations the best matches reveal a clear successional state of the forest for which simulated biomass values differ only slightly (Fig. 2b). Other locations, on the other hand, show ambiguities. Although matched simulated profiles look similar, simulated forest biomass values differ (Fig. 2a). We used the coefficient of variation of the probability distribution of forest attributes to define an uncertainty index $\varepsilon$ (Fig. 1; (3)). A regional analysis (Fig. 2c, d) reveals that the uncertainty index of AGB $\varepsilon_{AGB}$ is highest along the Arc of

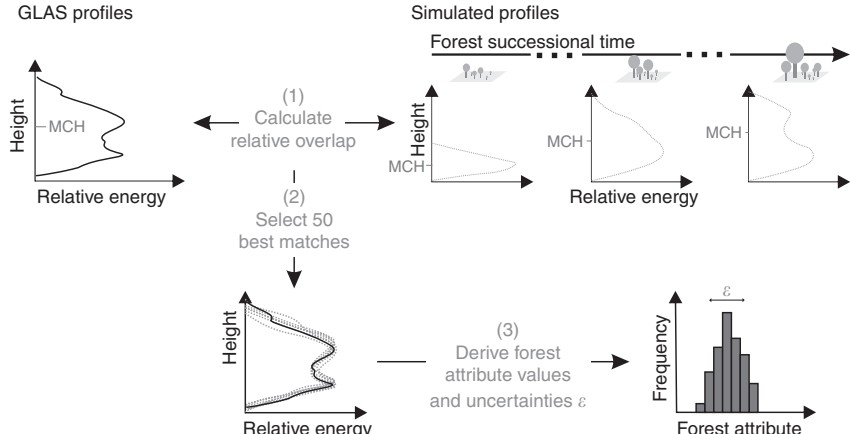

**Fig. 1** Workflow of our approach at one location in the Amazon. (1) The entire GLAS lidar profile derived from ICESat lidar is matched with simulated profiles (100 simulated lidar profiles at a 5-year interval over 2500 years of forest succession, 50,000 in total) by calculating their relative overlap. (2) We select the 50 best matches (relative overlap > 70%) between the GLAS profile and the simuletd profiles. (3) We then derive simulated forest attributes (e.g., biomass, basal area) for these 50 best matches. The derived probability distribution is used to quantify the uncertainty indices $\varepsilon$

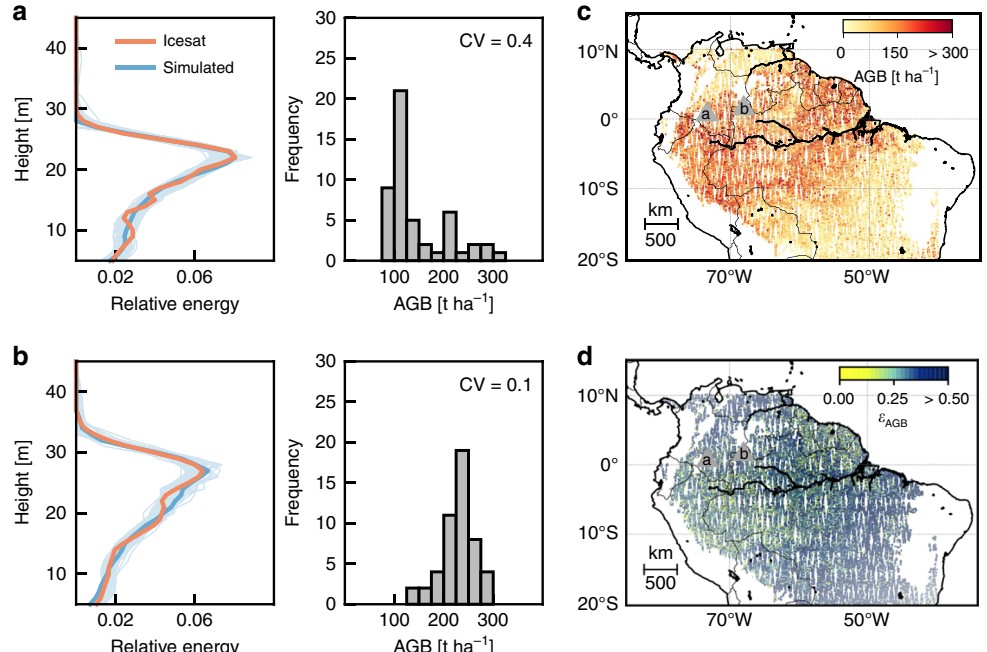

**Fig. 2** Two example locations where GLAS lidar profiles were matched with forest simulations. **a**, **b** left GLAS lidar profile (red) and best matching simulated profiles (blue for best relative overlap, light blue for the other 49 simulated profiles with highest overlap). The relative overlap of all profiles is >90%. **a**, **b** right Aboveground biomass (AGB) probability distribution for 50 simulated profiles and its coefficient of variation (CV). The CV is further denoted as $\varepsilon_{AGB}$, an uncertainty index that describes the structure- and species-induced uncertainty of AGB. **c** Mean AGB (gray triangles indicate the locations of the two example profiles.) and **d** $\varepsilon_{AGB}$-derived for every GLAS shot. Source data are provided as a Source Data file

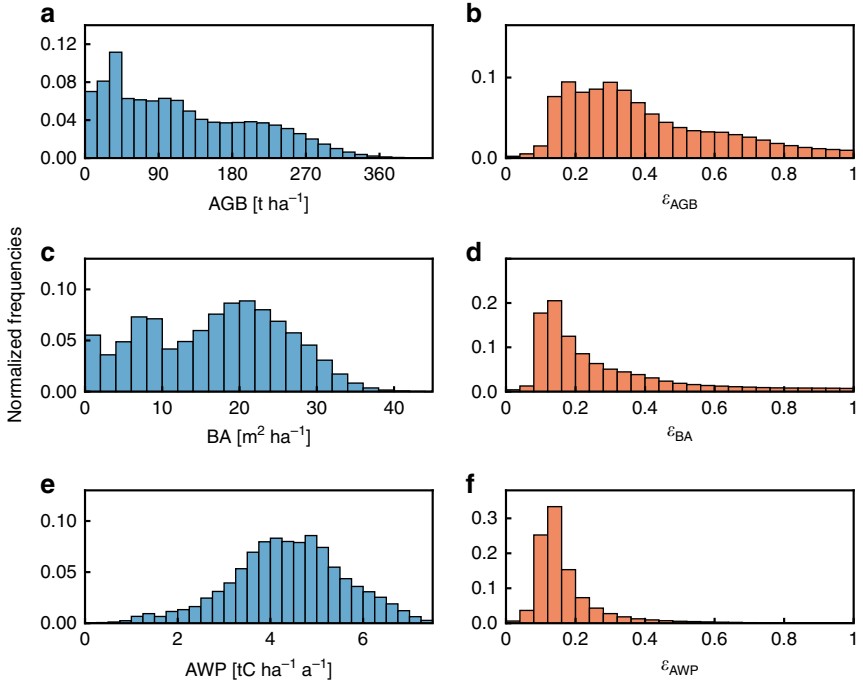

**Fig. 3** Frequency distributions of forest attributes and uncertainties. (left) Relative frequency distributions of **a** aboveground biomass (AGB), **c** basal area BA, and **e** aboveground wood productivity (AWP) for the Amazon based on 771,521 full lidar profiles. (right) Frequency distributions of the uncertainties for **b** biomass $\varepsilon_{AGB}$, **d** basal area $\varepsilon_{BA}$, and **f** aboveground wood productivity $\varepsilon_{AWP}$ (see Supplementary Fig. 1 for $\varepsilon_{SV}$, $\varepsilon_{GPP}$). Source data are provided as a Source Data file

Deforestation in the south–east, in central Amazon along the Amazon river and in the north–west. Ambiguousness of profiles has less influence on biomass values on the Guyana Shield and in the south–west toward the Andes.

**Amazon-wide forest attributes.** Our approach allows for deriving frequency distributions for AGB, BA, and AWP of the entire Amazon (Fig. 3, Supplementary Fig. 1 for SV and GPP). AGB values reach up to 520 t ha$^{-1}$ with a mean value of 120 t ha$^{-1}$, BA

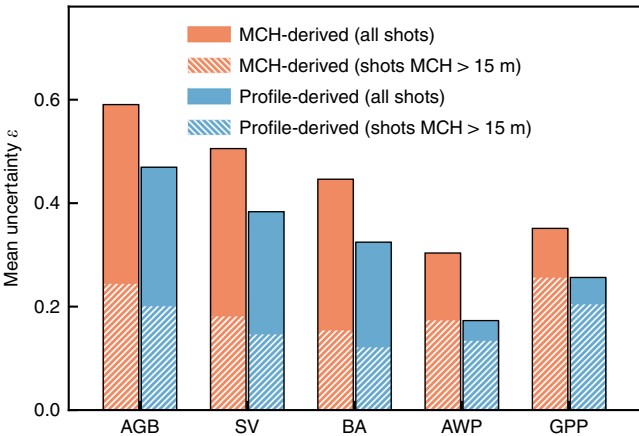

**Fig. 4** Mean uncertainties $\varepsilon$ for different forest attributes. Shown are $\varepsilon$ for aboveground biomass (AGB), stem volume (SV), basal area (BA), aboveground wood productivity (AWP), and gross primary productivity (GPP) taking mean canopy height (MCH) as a proxy for forest attributes (MCH-derived) vs. the entire lidar profile (profile-derived). Hatched bars show the mean uncertainty index for lidar profiles with a MCH > 15 m. Source data are provided as a Source Data file

up to 52 m² ha⁻¹ with a mean of 17 m² ha⁻¹, and AWP up to 8.7 tC ha⁻¹ a⁻¹ with a mean of 4.4 tC ha⁻¹ a⁻¹. The frequency distributions of the uncertainty of BA $\varepsilon_{BA}$ and of aboveground wood productivity (AWP) $\varepsilon_{AWP}$ have a clear defined peak at below 20%, whereas the distribution of $\varepsilon_{AGB}$ displays a wider span between 12 and 44%.

Further analyses show that the ambiguity in forest attributes decreases with increasing MCH of forests with the exception of AWP (Supplementary Fig. 2). Forests with a MCH < 15 m show highest ambiguity. Uncertainty $\varepsilon_{AGB}$ also decreases with increasing AGB while in terms of standard deviation (absolute values) it is more or less constant for all AGB values (Supplementary Fig. 3).

The mean uncertainty of AGB over all lidar profiles is ~47% (Fig. 4). However, when profiles with a MCH < 15 m are excluded, overall uncertainty is ~20%. Mean $\varepsilon_{BA}$ and $\varepsilon_{SV}$ are around and $\varepsilon_{GPP}$ is below 30%. $\varepsilon_{BA}$ decreases to 12% ($\varepsilon_{GPP}$ to 20%) for profiles with a MCH > 15 m. Uncertainty of AWP is below 20% for all profiles. We find that profile-derived values are ~20–43% more accurate than MCH-derived values, e.g., $\varepsilon_{BA}$ decreases from 45 to 32% and $\varepsilon_{AWP}$ from 30 to 17%.

## Discussion

The investigation across the Amazon shows that lidar profiles with a 65 m footprint can reveal carbon stocks and productivity of forests (MCH > 15 m) with a profile-matching uncertainty of ~20%. The derived uncertainty index of BA is even <15%.

Forest stands can include different tree species that differ in wood densities. For example, when a forest stand is dominated by pioneer species, biomass will be low because pioneer species tend to have low wood densities[33]. Thus, forest biomass values may differ although their tree size distributions and hence lidar profiles are the same (Fig. 2a). In the following, we refer to this effect as species-induced uncertainty. This effect alone, however, is not sufficient to explain the uncertainty in estimating biomass from lidar profiles. BA is a forest attribute that is independent of its species' wood densities but anyhow shows a mean uncertainty of ~ 15%. That means that a canopy profile can refer to different tree height distributions within the footprint, an effect which we call structure-induced uncertainty.

The quantified uncertainty of biomass is ~45% (20% for shots of MCH > 15 m), whereas the overall uncertainty of BA is ~3/4 of the uncertainty of biomass. As BA is the only forest attribute in this study that is independent of species wood densities and composition, this is a first indication that most of the profile ambiguity in biomass is structure-induced and only around 1/4 is species-induced (Fig. 4).

We find that the uncertainty of profile-derived biomass, BA, stem volume, and gross primary productivity decrease with increasing MCH (Fig. 4, Supplementary Fig. 2). This observation can be related to the variation of tree densities and gap dynamics in natural forests. Forest gaps show a high density of trees[34] so that the amount of possibilities to form a particular lidar profile is larger than for a few large trees. In addition, larger trees mainly occur in later successional state, which can only be reached by a few species. Consequently, forests of low canopy height reveal a higher structural- and species-induced ambiguity than forests that are only dominated by a few large trees.

The determined uncertainty of AWP is the lowest of all forest attributes (Fig. 4).The lidar profile quantifies how light penetrates into the forests. In other words, it reflects the amount of light that is absorbed by the forest. That means that AWP is mainly influenced by the forest light climate[35] and less by the forest's species or structure. Consequently, the information content of the entire profile is of central importance for the estimation of productivity. This results in a low uncertainty for AWP of ~ 17%. GPP, on the other hand, seems to be stronger controlled by the forest's species and structure.

The presented approach has several advantages. First, the full-profile approach encompasses a vast amount of forest states and reveals ambiguities of profiles. It thereby considers more forest stands than previous studies that often interpret remote sensing data based on statistical relations derived from a limited number of inventory plots (e.g.,[36]). This is of particular advantage in the Amazon region where the access of forests is limited and inventories of mature forest stands are often favored[37]. A comparison of inventory-based BA estimates with nearby GLAS-derived estimates (Supplementary Fig. 7) shows that the field inventory represents only one potential successional state while several GLAS shots reveal a highly heterogeneous forest. Empirical studies may benefit from this interpretation of lidar profiles and could take our quantification of uncertainties into account in the future. However, please note that our analysis here is performed for large footprints (65 m diameter). Uncertainties for such lidar footprints are potentially lower than those associated with smaller footprints owing to averaging effects[38].

Second, our approach allows for quantifying information content of entire lidar profiles. We have shown that values derived from an entire profile are up to 43% more accurate than when taking solely MCH as a proxy (Fig. 4). Thus, assimilating entire lidar waveforms is beneficial for estimates of actual AGB and productivity for large regions. For example, it reduces uncertainties of our previous approaches where sub-canopy structures were ignored and only aggregated values on 40 m × 40 m patches were considered[26,39].

Third, our approach is transferable to other spatial scales. Integrating forest models into the quantification of forest attributes from lidar will gain in importance with upcoming remote sensing missions. The GEDI mission, for example, will provide measurements for vegetation structure at a high resolution of 20 m[40]. We believe that taking the entire profile into account will reduce uncertainties related to structure-induced ambiguities. Species-induced ambiguities may be reduced by deriving leaf traits from future missions by hyperspectral measurements (e.g., EnMAP)[41].

Beside the uncertainties quantified here, estimates from lidar remote sensing have further sources of uncertainty: geolocation error, edge effects[42], instrument errors, errors in allometries[43]. Our uncertainty index ε quantifies structure- and species-induced uncertainties. It does not include geolocation errors that are mainly associated with inventory-based lidar studies (as in ref. [38]). Additional analyses have shown that the influence of changing the simulated footprint size (from 65 m to 60 m), and uncertainties that are related to the matching algorithm, are rather negligible (Supplementary Fig. 4). Note that proper uncertainty analyses on edge effects, footprint sizes, and tree allometries are limited by a high computational demand. Uncertainties of tree allometries may add another error for AGB of 10–20%[18].

Uncertainty in tree allometries is a common source of model structural and parameter uncertainties in dynamic vegetation models[44]. New approaches based on terrestrial laser scanning[45] will help to reduce allometric uncertainties in the future. However, in large-scale mapping projects that are based on spaceborne lidar, it remains a challenge to account for regional differences in allometries. As recently suggested, fusion of lidar remote sensing with forest models could contribute to improving our knowledge[46].

Our individual-based approach tackles the challenge of including spatial variable species dynamics. However, at the same time this complexity is limited by structural simplifications like categorizing tree species into mean early-, mid-, and late-successional plant functional types (PFT, aggregation approach). Model structural uncertainties are a matter of finding the right balance between simplification and complexity. A previous study[47] suggests to use at least one group that represents early successional tree species. Increasing the amount of PFTs by adding undergrowth shade-tolerant species could affect the derived uncertainty of AGB, GPP, and AWP, but at the same time this complexity relies on additional uncertain tree allometries of those tree species. A further simplification of our approach is that it does not consider the influence of nutrients on forest growth[48].

We conclude that forest modeling is a powerful tool to explore and quantify the information content of canopy profiles observed by lidar. Our approach complements inventory-based statistical approaches[6,49,50] by including a vast variety of forest successional states. Ecosystem modeling approaches have previously only used MCH as a proxy[24,26]. By making use of full lidar profiles, our approach advances the usage of lidar remote sensing for a high-resolution quantification of forest biomass and productivity.

## Methods

**Study area**. The study area covers forests in South America that are categorized as rainforest or moist deciduous rainforest (according to the FAO definition), have an annual mean temperature above 18 °C and are located at an elevation below 1000 m[26,51].

**Lidar data**. Lidar data have been derived from GLAS measurements on board ICESat between 2003 and 2006. We here use only lidar shots that fell into the Amazon region as defined above. Lidar shots have a footprint of ~65 m in diameter, an along-track distance of ~175 m and a between-track distance of ~30 km[30]. We used filtered data that exclude data of low quality (as in ref. [52]) and over steep slope (>10°), and that exclude leaf-off season derived from MODIS phenology. In order to reduce data volume, each energy profile was reconstructed from a set of Gaussian fitting parameters (the GLAS14 data product) to retrieve individual waveform[30]. In addition, we eliminated out all shots with a MCH shorter than 5 m. This results in 771,521 lidar shots in total.

**The Amazon-wide forest model**. The Amazon-wide version of the individual-based forest model FORMIND[26] was used to simulate forest dynamics throughout all successional states. It has been set up to reproduce stem size distributions, biomass, and BA at different successional states at four different locations in the Amazon region. Forest dynamics evolve from individual tree growth and

establishment, competition for light and space, and natural tree mortality[31,32]. Spatially variable insolation (photosynthetic photon flux density, PPFD) drives tree growth and competition for light, while precipitation and the clay fraction of soil drive individual tree mortality. AGB, SV, BA, GPP, and AWP can be analyzed at every successional state at different spatial resolutions (e.g., 0.16 ha, 1 ha, 1 km² in ref. [28]). The forest model has been validated for aboveground tree biomass and BA with forest inventory data[26] and cross-compared for productivity with other studies[39].

The Amazon rainforest was stratified into areas of similar environmental conditions to reduce computational effort: mean annual precipitation, clay fraction, and mean annual PPFD[26]. The forest model's input (PPFD, precipitation, clay fraction of soil) is variable in space, but constant in time. This resulted in 1280 areas in total for each of which 1 km² of forest succession, including individual trees with a diameter > 10 cm, was simulated representatively from bare ground over 2500 years. Approximately every 100 years, the simulated area was partly disturbed in order to cover all potential successional states as in ref. [53]. Thus, a representative simulation run was assigned to each 1-km² grid cell across South American tropical forests.

**Deriving waveform from the forest model**. For the simulation of the lidar waveforms we used the approach derived by Knapp et al. [53]. The forest stand is described as a three-dimensional voxel space, considering positions and crown dimensions of all trees. The voxels have a side length of 1 m and are filled representing the canopy. The reflected energy from each voxel is modeled as an exponential decay function (Beer-Lambert's law). To account for the characteristics of a GLAS footprint the following procedures were applied: a circular area of 65 m diameter[30] around the center of each simulated hectare was sampled. That means that we only use a subplot of each simulated hectare of the Amazon-wide forest model in order to consider the same spatial scale as a GLAS lidar shot. As the footprint can vary from 50 to 65 m varying from ellipsoid to circular[54], we additionally tested our approach with a 60 m footprint (Supplementary Fig. 4). The contribution of each voxel to reflection was weighted based on the horizontal distance to the pulse center using a 2-dimensional Gaussian function (SD = 1/4 × diameter of footprint) resulting in full contribution in the center and a decreased contribution at the edge. Energy of all voxels per 1-m height layer was added up and the derived waveform was finally normalized by the total sum to obtain relative energy per height layer. For each of the 1280 areas, we simulated one hundred footprints of 65 m radius over 2500 years at a 5-year time step. These sum up to 64 million simulated footprints in total. Note that in the current version, the forest model simulates forest succession under stable climatic condition on yearly time steps. Hence, our approach does not consider any intra-annual changes in canopy profiles as observed in lidar profiles by Tang & Dubayah[30].

**Identifying successional states with lidar profiles**. Each ICESat lidar profile was compared with simulated lidar profiles of local forest succession (100 representative simulated footprints over 2500 years). The similarities of normalized (by total area including ground return) observed and simulated profiles were determined by quantifying their relative overlap above 5 m (intersection area divided by union area, Eq. (1)):

$$O_{rel} = \sum_{h=5}^{50} \frac{\min(E_{obs}(h), E_{sim}(h))}{\max(E_{obs}(h), E_{sim}(h))} \tag{1}$$

where $O_{rel}$ is the relative overlap of the lidar observed and simulated profile. $E_{obs}$ is the lidar observed relative energy at height h, and $E_{sim}$ is the simulated relative energy at height $h$. We here assumed height classes of 1 m.

An analysis has shown that we can find a best relative overlap of >70% for 93% of all GLAS shots (Supplementary Fig. 5b; example for a 70% overlap in Supplementary Fig. 6; relationship between overlap and uncertainty see Supplementary Fig. 9). Consequently, we took the 50 best matching simulated profiles with an overlap above the threshold of 70% (Fig. 2 left) to identify potential successional states of the forest and their attributes, e.g., aboveground biomass. Supplementary Fig. 5a demonstrates the sensitivity of the uncertainty index to taking 50 samples. An upper limit needed to be set in order to handle forests in different successional stages equally as old growth forests occurred more frequent in our simulations than early successional stages. In order to account for uncertainties of the matching algorithm, the approach was additionally tested for a threshold of 60 and 80% (Supplementary Fig. 4a). If the relative overlap with a simulated profile was below the threshold, only the three best matches were taken. For each GLAS lidar profile, we hence derived up to 50 matching forest attribute values from our forest model. The distinctness or uncertainty of a forest attribute is then expressed as its coefficient of variation (CV) of all these values (Fig. 1; (3)). This CV is defined as the uncertainty index ε (see Supplementary Figs. 10 and 11 for an additional uncertainty (quartile coefficient of dispersion) that was tested in the framework of this study). Beside the profile-derived uncertainty index, we additionally determined the ambiguity of forests states of the 50 best matches when taking solely one height metric, here the MCH.

We tested the validity of the approach by comparing observed BA at 140 inventory plots[20,55] against simulated values derived from the nearest lidar shots (Supplementary Figs. 7 and 8). We consider all lidar shots that are located in a

distance smaller than 3 km around the inventory as the locations of the lidar shots do not match the exact locations of inventory data and coordinates of the inventory sites come with uncertainties[18].

**Reporting summary**. Further information on research design is available in the Nature Research Reporting Summary linked to this article.

## Data availability

Source data underlying Figs. 2, 3, 4, and Supplementary Figures are provided as a Source Data file.

## Code availability

The FORMIND model is freely available on http://formind.org/downloads/.

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

## Acknowledgements

This study was supported by the Helmholtz-Alliance Remote Sensing and Earth System Dynamics HA-310 under the funding reference RA37012. NK was supported by the German Federal Ministry for Economic Affairs and Energy (BMWi) under the funding reference 50EE1416. We acknowledge the efforts of the TEAM, RAINFOR, and ATDN projects making inventory data sets available. We also thank Michael Müller for technical support. R.D. and H.T. were supported by a NASA contract to the University of Maryland for the Global Ecosystem Dynamics Investigation (#NNL15AA03C) in addition to support from NASA's New Investigator Program (#80NSSC18K0708) to H.T.

## Author contributions

E.R., N.K., R.F., and A.H. conceived the study. E.R. and A.H. designed the study. E.R., N.K., and H.T. processed and analyzed data. E.R., N.K., R.F., F.B., R.D., H.T., and A.H. commented on analyses and contributed in writing the paper.

## Competing interests

The authors declare no competing interests.
