## [Peer Review File · Nature Communications]

Reviewers' comments:

Reviewer #2 (Remarks to the Author):

Dear editor,

This manuscript is an important study on a highly relevant topic in environmental research: how can we estimate and predict the carbon sequestration of forests as accurately as possible assuming that we can use remote sensing data only?

The particular challenges of the work result from the chosen study area (Amazon Basin): (1) The study area is immensely large. (2) The species diversity of the trees is large. (3) The factors influencing tree growth (soil quality, topography, water availability, etc.) are heterogeneous. (3) The entire Amazon Basin consists of a mosaic of forest patches that are in different succession stages. (4) Only a few inventory data are available for areas that are not systematically distributed over the area; and it must even be assumed that not all inventory data are representative if, for example, the areas were selected according to accessibility.

Although LIDAR techniques have been developed to collect 3D information on forest structure, the problem of accurately assessing forest carbon storage, productivity and structural development is still unsolved. This is mainly due to the fact that the interpretation of LIDAR data requires information on statistical relationships (e.g. tree height and biomass), which are only available from detailed and systematic inventory data that are missing there.

The authors solve this problem by a newly developed combination of LIDAR measurements (they use, in particular, the Geoscience Laser Altimeter System GLAS) with simulation experiments performed with the individual-based model FORMIND. The latter simulates the establishment, growth, and mortality of trees on spatial patches (20 m × 20 m). Competition for light is a main (but not the only driver) of these processes. To deal with the high level of biodiversity, the model groups tree species with similar ecological traits (e.g., their appearance in the succession cycle, their typical position in the different canopy layers) into plant functional types (PFTs).

To close the missing link between the GLAS measurements and the 3D structure of the forest, the authors now use the individual-based architecture of FORMIND. In this study, the 3D structure of the simulated forest patches is translated into vertical profiles of relative energy, which then can be directly compared with the profiles obtained by GLAS. The project idea is then straightforward but brilliant: (1) FORMIND is used to simulate the succession of forest patches. (2) the profiles of relative energy obtained by GLAS and FORMIND simulations will be compared in order to filter the relevant succession stage specifying the relevant 3D structure of the patch. (3) the forest parameters of interest (above-ground biomass, productivity, basal area etc.) will be extracted from FORMIND output and thus connected to the GLAS data.

This workflow is reasonable and well described. It is a welcome and thoughtful way to link small-scale forest structure with carbon estimates, and could indeed improve our understanding about the Amazon-wide biomass storage. Nevertheless, there are some issues, which I cannot accurately evaluate based on the present version of the manuscript:

1. The accurateness of the estimate strongly depends on the accurateness of the FORMIND simulation for the whole Amazon basin. This approval was not provided in the frame of this manuscript but elsewhere. This is o.k., but it should be discussed that the accuracy of the model is key for the quality of the results presented here. For example, it was stated in Knapp et al. 2018 that "the structural validity of the simulated old growth stands was confirmed by visually comparing biomass stocks (Fig. S1) and stem size distributions (Fig. S2) of all PFTs to the values obtained from the inventory data". Are the inventory data sufficiently representative for the whole Amazon basin?

2. I was checking the paper Rödiger et al. 2017 which provides a good overview about the upscaling of FORMIND from local forest dynamics over regional scales up to the Amazon region. The authors mention and quantify several uncertainties of FORMIND here (e.g., the tree height estimate of FORMIND, basal area – biomass, allometric changes etc.). How do these uncertainties impact the results presented in the ms to hand? Also, it would be worth to mention some more details of the amazon-version of FORMIND here, which are relevant to understand the approach. For example, at which patch size was the energy profile calculated 40 m x 40 m (Rödiger et al. 2017) or 60 m x 65 m (named as LIDAR food print size here)?

3. The authors calculate the level of uncertainty of their estimates based on the comparison between the GLAS profiles and FORMIND files only. It would be good to include the two points mentioned above in the uncertainty discussion. This is important because, we are finally interested in the accurateness of the “real” biomass estimate and not on the accurateness of the estimate of the simulated biomass.

Minor comments

1. Although the ms presents a new approach that will significantly improve the biomass estimates of forests, the text is yet not strong enough to highlight the innovative character of the study. The abstract provides correct information, but is (in my opinion) not sufficiently attractive to target on the audience of Nature Communication.

2. The same belongs to the introduction (particularly the first paragraph). It contains unnecessary information about older remote sensing techniques, but does not explain well why we need to improve the link between 3D forest structure and biomass estimates of forest. The hint that 3D information, tree heights, and canopy levels are important forest characteristics to be determined appears much too late in the text.

3. I am afraid that the used coefficient of variance is not the most suitable index for uncertainty, particularly, if the distributions are not symmetric and have several peaks. May be the quartile coefficient of dispersion is more suitable?

In conclusion, this ms presents a very important approach that is relevant for solving several recent challenges in environmental sciences namely the accurate estimate of forest biomass on regional level. It was applied for the Amazon region, which is of particular interest considering the ongoing trend of forest degradation here and its impact on global processes such as carbon cycle, climate dynamics etc. It is noteworthy to mention, that the combination of individual-based modelling with LIDAR measurements presented here, goes far beyond its application for the Amazon region. I am sure that it can serve as a roadmap for further studies done by other scientists for other systems, with other models, and for other regions. Two issues, however, have to be addressed by the authors before the ms can be published: (1) the text has to be polished so that the innovative character of the approach comes to a glance, and (2) the discussion of the uncertainties of the results related to the FORMIND model itself has to be improved.

Reviewer #3 (Remarks to the Author):

The manuscript by Rodig et al demonstrates how a forest model can be used with waveform lidar to estimate forest biomass. The authors have developed a neat approach whereby a forest gap model is used to simulate lidar waveforms. These simulated waveforms are then compared with observed waveforms from ICESAT-1 GLAS instrument. The waveforms that match are used to derive various forest states, i.e., biomass, basal area etc., and the uncertainty is derived from the ensemble of potential matches between the GLAS waveform and simulated waveform.

The manuscript is well written and I have only minor comments.

The main text needs to present in more detail how the waveform 'match' procedure is made. This is a critical piece of the work and I think it deserves to be in the main text with an equation if possible.

Why are the top 50 matches kept and the rest removed? What is the sensitivity of the results to this assumption?

Lastly, is the uncertainty estimated from using only the 'ensemble' of simulated waveforms that closely match GLAS? I would think that the in situ forest plot data would be the final check of uncertainty. Please can you clarify and reconsider how the statistics are generated.

Reviewer #2 (Remarks to the Author):

Dear editor,

This manuscript is an important study on a highly relevant topic in environmental research: how can we estimate and predict the carbon sequestration of forests as accurately as possible assuming that we can use remote sensing data only?

The particular challenges of the work result from the chosen study area (Amazon Basin): (1) The study area is immensely large. (2) The species diversity of the trees is large. (3) The factors influencing tree growth (soil quality, topography, water availability, etc.) are heterogeneous. (3) The entire Amazon Basin consists of a mosaic of forest patches that are in different succession stages. (4) Only a few inventory data are available for areas that are not systematically distributed over the area; and it must even be assumed that not all inventory data are representative if, for example, the areas were selected according to accessibility.

Although LIDAR techniques have been developed to collect 3D information on forest structure, the problem of accurately assessing forest carbon storage, productivity and structural development is still unsolved. This is mainly due to the fact that the interpretation of LIDAR data requires information on statistical relationships (e.g. tree height and biomass), which are only available from detailed and systematic inventory data that are missing there.

The authors solve this problem by a newly developed combination of LIDAR measurements (they use, in particular, the Geoscience Laser Altimeter System GLAS) with simulation experiments performed with the individual-based model FORMIND. The latter simulates the establishment, growth, and mortality of trees on spatial patches (20 m × 20 m). Competition for light is a main (but not the only driver) of these processes. To deal with the high level of biodiversity, the model groups tree species with similar ecological traits (e.g., their appearance in the succession cycle, their typical position in the different canopy layers) into plant functional types (PFTs).

To close the missing link between the GLAS measurements and the 3D structure of the forest, the authors now use the individual-based architecture of FORMIND. In this study, the 3D structure of the simulated forest patches is translated into vertical profiles of relative energy, which then can be directly compared with the profiles obtained by GLAS. The project idea is then straightforward but brilliant: (1) FORMIND is used to simulate the succession of forest patches. (2) the profiles of relative energy obtained by GLAS and FORMIND simulations will be compared in order to filter the relevant succession stage specifying the relevant 3D structure of the patch. (3) the forest parameters of interest (above-ground biomass, productivity, basal area etc.) will be extracted from FORMIND output and thus connected to the GLAS data.

Response: Thank you very much for your supportive comments.

This workflow is reasonable and well described. It is a welcome and thoughtful way to link small-scale forest structure with carbon estimates, and could indeed improve our understanding about the Amazon-wide biomass storage. Nevertheless, there are some issues, which I cannot accurately evaluate based on the present version of the manuscript:

1. The accurateness of the estimate strongly depends on the accurateness of the FORMIND simulation for the whole Amazon basin. This approval was not provided in the frame of this manuscript but elsewhere. This is o.k., but it should be discussed that the accuracy of the model is key for the quality of the results presented here. For example, it was stated in Knapp et al. 2018 that “the structural validity of the simulated old growth stands was confirmed by visually comparing biomass stocks (Fig. S1) and stem size distributions (Fig. S2) of all PFTs to the values obtained from the inventory data”. Are the inventory data sufficiently representative for the whole Amazon basin?

Response: Thank you for noticing. We have added some remarks regarding the accuracy of the model to the methods section (line 254-255).

Yes, the accuracy of the model was discussed in Rödiger et al. 2017 (GEB) for different forest stands in different successional states. The forest model was fine-tuned against 4 forest sites in climax state and 10 sites in earlier successional states spread throughout four different locations in the Amazon region (tree records of the Large Scale Biosphere-Atmosphere Experiment in Amazonia (LBA):

Fig. S2 in Rödiger et al, 2017, GEB: (a-d) Simulation of forest succession at different sites (locations listed in Tab. S 3) for three different plant functional types (early, mid and late successional trees) using the best fit parameters (Tab. S 2). The simulation envelope shows the variation of simulated biomass at the scale of 1 ha (95% quantile for 100 simulations). Comparison with field data (dots) in different successional stages (different ages of forest stands at YA, BR and PP recorded in Brondizio & Moran, 2009; at Manaus (MA) we assume an old-growth forest in climax stage). (e-k) Simulated tree size distributions compared to field data in different successional stages. The simulation envelopes show the variations at the scale of 1 ha (95% quantile for 100 simulations).

For the validation of aggregated biomass and basal area values, we have used 114 field inventories at different successional states throughout the Amazon. It is difficult to state whether the inventory data sufficiently representative for the whole Amazon basin. As you have mentioned yourself above, “Only a few inventory data are available for areas that are not systematically distributed over the area; and it must even be assumed that not all inventory data are representative if, for example, the areas were selected according to accessibility.”

2. I was checking the paper Rödiger et al. 2017 which provides a good overview about the upscaling of FORMIND from local forest dynamics over regional scales up to the Amazon region. The authors mention and quantify several uncertainties of FORMIND here (e.g., the tree height estimate of FORMIND, basal area – biomass, allometric changes etc.). How do these uncertainties impact the results presented in the ms to hand? Also, it would be worth to mention some more details of the

amazon-version of FORMIND here, which are relevant to understand the approach. For example, at which patch size was the energy profile calculated 40 m x 40 m (Rödig et al. 2017) or 60 m x 65 m (named as LIDAR food print size here)?

Response: Thank you for your comment. We have added more information about the patch sizes to the introduction (line 72-73) and the differences to our previous approach to the methods section (line 278-280).

We have expanded the discussion (line 197-199) on uncertainties regarding tree height vs. full profile in order to highlight the advances of the study in hand.

Also, we now discuss uncertainties related to tree allometries in more detail. Tree allometries can be a source of uncertainties for biomass estimates. These uncertainties, however, affect studies that are based on individual-based forest modelling, as well as in inventory-based studies. For that reason, we have based the validation of our approach on a comparison with observed basal area (as basal area is not affected by uncertainties in tree allometries, Fig. S 7). We have added this aspect to the discussion (line 217-222).

3. The authors calculate the level of uncertainty of their estimates based on the comparison between the GLAS profiles and FORMIND files only. It would be good to include the two points mentioned above in the uncertainty discussion. This is important because, we are finally interested in the accurateness of the “real” biomass estimate and not on the accurateness of the estimate of the simulated biomass.

Response: Thank you for your comment. We have carefully revised the uncertainty discussion in order to highlight how accurateness influences biomass estimates (line 217-232).

Minor comments

1. Although the ms presents a new approach that will significantly improve the biomass estimates of forests, the text is yet not strong enough to highlight the innovative character of the study. The abstract provides correct information, but is (in my opinion) not sufficiently attractive to target on the audience of Nature Communication.

Response: Thank you for your comment. We have revised the abstract carefully. Your summary of the manuscript has helped us a lot to improve the abstract. We hope that we now highlight more the innovative character of this study.

2. The same belongs to the introduction (particularly the first paragraph). It contains unnecessary information about older remote sensing techniques, but does not explain well why we need to improve the link between 3D forest structure and biomass estimates of forest. The hint that 3D information, tree heights, and canopy levels are important forest characteristics to be determined appears much too late in the text.

Response: Thanks for this comment. We have changed the first paragraph, now highlighting the importance of 3D information in the assessment of carbon stocks and fluxes.

3. I am afraid that the used coefficient of variance is not the most suitable index for uncertainty, particularly, if the distributions are not symmetric and have several peaks. May be the quartile coefficient of dispersion is more suitable?

Response: Thank you for this comment. We have tested the quartile coefficient of dispersion (QC) for all forest attributes. Our analysis shows that the QC value is generally lower than the coefficient of variation (CV). We had expected such a result as the QC excludes outliers (Fig. S 10).

Interestingly, the profile-derived values are by the same amount more accurate than MCH-derived values as when taking the CV as uncertainty index: QC_{AGB} by 21%, QC_{SV} by 22%, QC_{BA} by 24%, QC_{AWP} by 44%, and QC_{GPP} by 25% (Fig. S11). As the CV is a more commonly used index for uncertainty, we prefer to stick to the original uncertainty index. Anyhow, we have added the results of our additional study to the supplements (Fig. S10 and 11) and added a reference to the method section (line 315-316).

In conclusion, this ms presents a very important approach that is relevant for solving several recent challenges in environmental sciences namely the accurate estimate of forest biomass on regional level. It was applied for the Amazon region, which is of particular interest considering the ongoing trend of forest degradation here and its impact on global processes such as carbon cycle, climate dynamics etc. It is noteworthy to mention, that the combination of individual-based modelling with LIDAR measurements presented here, goes far beyond its application for the Amazon region. I am sure that it can serve as a roadmap for further studies done by other scientists for other systems, with other models, and for other regions. Two issues, however, have to be addressed by the authors before the ms can be published: (1) the text has to be polished so that the innovative character of the approach comes to a glance, and (2) the discussion of the uncertainties of the results related to the FORMIND model itself has to be improved.

Response: Thank you very much for your supportive comments. With the help of the review, we could improve our abstract and introduction by highlighting the novelty of our study. We have also extended the uncertainty discussion.

Reviewer #3 (Remarks to the Author):

The manuscript by Rodig et al demonstrates how a forest model can be used with waveform lidar to estimate forest biomass. The authors have developed a neat approach whereby a forest gap model is used to simulate lidar waveforms. These simulated waveforms are then compared with observed waveforms from ICESAT-1 GLAS instrument. The waveforms that match are used to derive various forest states, i.e., biomass, basal area etc., and the uncertainty is derived from the ensemble of potential matches between the GLAS waveform and simulated waveform.

The manuscript is well written and I have only minor comments.

Response: Thank you.

The main text needs to present in more detail how the waveform ‘match’ procedure is made. This is a critical piece of the work and I think it deserves to be in the main text with an equation if possible.

Response: We agree with the reviewer. We have added additional explanations and the equation to the main text (line 297-300). Thank you for noticing.

Why are the top 50 matches kept and the rest removed? What is the sensitivity of the results to this assumption?

Response: Thank you for your comment. We have run several tests to investigate how many simulated profiles are needed to cover an entire range of forest structures. Fig. S5 demonstrates the sensitivity of the uncertainty index to this assumption. The distribution of the uncertainty index barely changes above 25 samples. We needed to set an upper limit in order to handle forests in different successional stages equally as old growth forests occurred more frequent in our simulations than early successional stages. We have added an explanation to the main text (line 305-308).

Lastly, is the uncertainty estimated from using only the ‘ensemble’ of simulated waveforms that closely match GLAS? I would think that the in situ forest plot data would be the final check of uncertainty. Please can you clarify and reconsider how the statistics are generated.

Response: Thank you for raising this important issue. That is why we have done a final check with in situ forest plot data (Fig. S7). We realized that we have to make this point clearer and added a remark to the discussion (line 186-189). Using in situ plot data as a final check is a challenge, as the locations of the lidar shots do not match the exact locations of inventory data. We here used lidar shots that fell into a 3km radius of a forest inventory in the Amazon. Considering the fact that the coordinates of the inventory sites come with uncertainties and that forest patches within a 60m footprint can be highly heterogeneous (studies have shown that at least 4 ha (Réjou-Méchain et al. (2014), Biogeosciences) to 10 ha (Chambers et al. (2013), PNAS) of inventory are needed to cover all local spatial variability), we believe that the presented validation is reasonable. We have added an additional figure to the supplements (Fig. S8). This figure displays the potential of lidar measurements to capture the full heterogeneity of a forest; as compared to inventory measurements that (understandably) only represent fractions of the forest with a bias towards mature forest stands.

REVIEWERS' COMMENTS:

Reviewer #2 (Remarks to the Author):

The authors addressed all my comments given to the previous version of the manuscript. I do not have further questions or issues to be improved.

Reviewer #3 (Remarks to the Author):

Thank you for addressing my minor comments. This is an important study and presentation of novel techniques and I am very pleased to see the revisions and improved clarity.

REVIEWERS' COMMENTS:

Reviewer #2 (Remarks to the Author):

The authors addressed all my comments given to the previous version of the manuscript. I do not have further questions or issues to be improved.

Thank you very much for your helpful comments.

Reviewer #3 (Remarks to the Author):

Thank you for addressing my minor comments. This is an important study and presentation of novel techniques and I am very pleased to see the revisions and improved clarity.

We thank you for your very supportive feedback.